# Age influences on Propofol estimated brain concentration and entropy during maintenance and at return of consciousness during total intravenous anesthesia with target-controlled infusion in unparalyzed patients: An observational prospective trial

Federico Linassi[1,2]*, Matthias Kreuzer[2], Eleonora Maran[1], Antonio Farnia[3], Paolo Zanatta[4], Paolo Navalesi[1], Michele Carron[1]

1 Department of Medicine—DIMED, Section of Anesthesiology and Intensive Care, University of Padova, Padova, Italy, 2 Department of Anesthesiology and Intensive Care, Klinikum rechts der Isar, Technical University of Munich, School of Medicine, Munich, Germany, 3 Department of Anesthesia and Intensive Care, Treviso Regional Hospital, Piazzale Ospedale, Treviso, Italy, 4 Department of Anesthesia and Intensive Care, Integrated University Hospital of Verona, Piazzale Aristide Stefani, Verona, Italy

* federico.linassi@gmail.com

## Abstract

### Purpose

Aging affects pharmacodynamics/pharmacokinetics of anesthetics, but age effects on Entropy-guided total intravenous anesthesia with target-controlled infusions (TIVA-TCI) are not fully characterized. We compared aging effects on effective estimated brain concentration of Propofol (CeP) during TIVA-TCI Entropy-guided anesthesia, without neuromuscular blockade (NMB).

### Methods

We performed an observational, prospective, single-center study enrolling 75 adult women undergoing Entropy-guided Propofol-Remifentanil TIVA-TCI for breast surgery. Primary endpoint was the relationship between age and CeP at maintenance of anesthesia (MA) during Entropy-guided anesthesia. Secondary endpoints were relationships between age and CeP at arousal reaction (AR), return of consciousness (ROC) and explicit recall evenience. We calculated a linear model to evaluate the age's impact on observational variable and performed pairwise tests to compare old (≥65 years, n = 50) and young (<65 years, n = 25) patients or patients with and without an AR.

### Results

We did not observe age-related differences in CeP during MA, but CeP significantly (p = 0,01) decreased with age at ROC. Entropy values during MA increased with age and were

**Data Availability Statement:** All relevant data are within the manuscript and its Supporting Information files.

**Funding:** Authors received no specific funding for this work

**Competing interests:** The authors declare that no competing interests exists

**Abbreviations:** ASA, American Society of Anesthesiolgists; BIS, Bispectral Index (Medtronic, Dublin, Ireland); BP, Blood Pressure; BSupp, Burst suppression; CeP, estimated brain concentration of Propofol; CeR, estimated brain concentration of Remifentanil; CI, Confidence Interval; EMG, electromyography; IQR, Interquartile range; LMA, laryngeal mask airway; LOC, loss of consciousness; MA, maintenance of anesthesia; NMB, neuromuscular blocking agents; pEEG, processed electroencephalographic; PK/PD, pharmacokinetics/pharmacodynamic; POCD, Post-Operative Cognitive Dysfunction; RE, Response Entropy (Entropy Module, GE Healthcare, Chicago, IL, USA); ROC, return of consciousness; SD, Standard deviation; SE, State Entropy (Entropy Module, GE Healthcare, Chicago, IL, USA); SPI, Surgical Pleth Index; TIVA-TCI, Total intravenous anesthesia with target-controlled infusions.

significantly higher in the elderly (RE: median 56[IQR49.3–61] vs 47.5[42–52.5],p = 0.001; SE: 51.6[45–55.5] vs 44[IQR40-50],p = 0.005). 18 patients had an AR, having higher maximum RE (92.5[78–96.3] vs 65[56.5–80.5],p<0.001), SE (79[64.8–84] vs 61[52.5–69],p = 0.03, RE-SE (12.5[9.5–16.5] vs 6 [3–9],p<0.001.

## Conclusion

Older age was associated with lower CeP at ROC, but not during MA in unparalysed patients undergoing breast surgery. Although RE and SE during MA, at comparable CeP, were higher in the elderly, Entropy, and in particular an increasing RE-SE, is a reliable index to detect an AR.

## Introduction

Total intravenous anesthesia with target-controlled infusions (TIVA-TCI) allows anesthetists to achieve a stable plasma or estimated brain concentration of Propofol and Remifentanil (CeP and CeR, respectively) and to promptly respond to signs of inappropriate anesthetic/analgesic plans, being considered the "ideal" approach by some clinicians [1]. TIVA-TCI systems use multi-compartment pharmacodynamics/pharmacokinetic models to calculate the necessary infusion rates to reach and maintain CeP and CeR, based on patient age, sex, weight, and height [2–5]. CeP at loss of consciousness (LOC) and return of consciousness (ROC) during TIVA-TCI are lower in the elderly (age ≥65 years) than in younger patients (age <65 years) [6,7]. Further, it seems that lower CeP is necessary to maintain a similar anesthetic level in elderly using TIVA-TCI during cardiac surgery [8], suggesting that older patients are more sensitive to Propofol administered for anesthesia maintenance.

Commercial monitors using processed electroencephalographic (pEEG) parameters can be used to target the adequate range of general anesthesia, by titrating the anesthetic to a recommended index range. The Entropy module calculating the State (SE) and Response Entropy (RE) (Entropy Module, GE Healthcare, Chicago, IL, USA) [9] is one of these devices that translates the EEG activity into dimensionless indices inversely related to the anesthetic level by using specially designed algorithms [9]. The use of these indices seems to decrease the risk of Post-Operative Cognitive Dysfunction (POCD) or intraoperative awareness by helping to avoid excessively deep or light levels of anesthesia [10,11], but their usefulness for these purposes is matter of controversial discussion [12–15]. Previous results describe an influence of age on these indices [16,17], because aging changes intraoperative EEG features [18,19].

Noxious stimulation during general anesthesia can cause arousal events that are accompanied by body movements in the absence of neuromuscular blocking agents. These events may reflect an increased probability of consciousness or insufficient general anesthesia without neuromuscular blockade [20]. To detect arousal reactions, non EEG-based parameters such as the Surgical Pleth Index (SPI; GE Healthcare, Helsinki, Finland) also exist. The SPI measures the nociception/anti-nociception balance during anesthesia and is a function of stimulation intensity and the analgesic component provided by an opiate infusion [21]. An increase in SPI or RE and RE-SE is indicative of an arousal reaction with movement [22]. Here, we present results from our investigation regarding the impact of age on pEEG-guided TIVA-TCI as well as the RE, SE and SPI reaction during intraoperative arousal events in the absence of neuromuscular blockade.

## Methods

This observational, prospective study was approved by the Ethical Committee of Treviso Regional Hospital, Italy (N. 681/CE Marca) and registered with ClinicalTrials.gov (NCT04129112). All procedures in the study were in accordance with the ethical standards of our institutional and/or national research committees, as well as the 1964 Helsinki Declaration and its later amendments or comparable ethical standards.

After obtaining written informed consent, we recruited adult (age ≥18 years) females undergoing oncologic breast surgery (quadrantectomy or mastectomy) at the Treviso Regional Hospital from July 1, 2019 to December 15, 2019. We excluded patients with neurological, cerebrovascular, or psychiatric disease, with severe respiratory, cardiovascular, kidney, or liver disease, or patients with American Society of Anesthesiolgists (ASA) classification > II. Patients in continuous therapy with anti-depressive drugs, anxiolytics, or with a history of drugs or psychoactive drug abuse were excluded. We also excluded patients if they required preoperative anxiolysis, intraoperative vasoactive drugs for hemodynamic instability, or received neuromuscular blocking agents (NMB). S1 Fig presents the corresponding CONSORT chart.

### 2.1 General anesthesia

Routine monitoring included continuous electrocardiogram, pulse-oximetry, and non-invasive blood pressure (BP) measurements. We placed an EEG electrode strip on the patient's forehead to monitor the brain electrical activity with the Entropy Module (GE Entropy™ Module, GE Healthcare). RE (display range, 0–100) and SE (display range, 0–91) were continuously monitored and displayed. SE reflects the spectral Entropy of the EEG signal calculated up to 32 Hz, whereas RE is the spectral Entropy including higher frequencies (up to 47 Hz) that includes the electromyography- (EMG)-dominant part of the spectrum [9]. Consequently, frontal EMG activity causes a fast response of RE and an increase in RE-SE. We used the SPI to monitor hemodynamic responses to surgical stimuli [22]. The unique SPI algorithm processes plethysmographic amplitude and pulse intervals to create a single index. SPI between 20 and 50 seem to present target values for Propofol-Remifentanil anesthesia [23].

We used A TIVA-TCI pump (Orchestra® Base Primea with two module DPS, Fresenius-Kabi, Brézins, France) to induce and maintain general anesthesia, targeting the CeP using the Schnider model [2,3] and the CeR using the Minto model [4,5]. According to good standard for routine clinical practice, we suggested anesthesiologist to target CeP values to the range of 2–4 μg ml$^{-1}$ [1], then adjusted to achieve a target SE of 40–60 [25]. When SE deviated from this range, CeP was adjusted in 0.5 μg ml$^{-1}$ increments at intervals of ≥1 min until SE returned to the suggested range. To avoid side effects (e.g., tachycardia, hypotension, and chest wall rigidity), fentanyl 1–2 μg kg$^{-1}$ was administered to facilitate laryngeal mask airway (LMA) insertion. CeR was increased in 0.5 ng ml$^{-1}$ increments every 2 min until the suggested range of 2–3 ng ml$^{-1}$ [1], was reached. CeR was then adjusted to a target SPI between 20 and 50. When the SPI deviated from this range, CeR was adjusted in 0.5 μg kg$^{-1}$ increments at intervals of ≥1 min until the SPI returned to the suggested range [23].

We gently placed a LMA following LOC and ventilated the lungs (Primus Anesthesia Workstation, Drager, Telford, PA, USA) at a tidal volume of 6–8 ml kg$^{-1}$ using volume-control mode with an inspiration:expiration ratio of 1:2 and positive end-expiratory pressure of 5 cm $H_2O$. The respiratory rate was initially 12 min$^{-1}$ and subsequently adjusted to maintain an end-tidal $CO_2$ of 35 to 40 mmHg.

After surgery, we targeted the TIVA-TCI to a CeP of 0 μg ml$^{-1}$ and CeR of 0 ng ml$^{-1}$. We removed the LMA at ROC, defined as spontaneous eye opening and execution of simple commands.

## 2.2 Clinical endpoints and variables

The aim of this study was to assess the effects of age on Entropy-guided TIVA-TCI anesthesia. Primary end-point was the relationship between age and TIVA-TCI's *effective* estimated brain concentrations of Propofol (CeP) and Remifentanil (CeR) during anesthesia maintenance.

Secondary endpoints were the relationship between age and CeP at arousal, ROC and the occurrence of explicit recall.

For the primary endpoint, the following potential confounder variables were considered: body mass index (BMI), ASA physical status, number of previous general anesthetic exposure, years of schooling, surgery duration, fentanyl use (total quantity).

An anesthetist among the authors, who was not involved in delivering anesthesia to patients, recorded variables on a paper data-collection form and was in charge to record any arousal events, defined as *any* involuntary movement, inadequate ventilation because of vocal cord closure, or significant hemodynamic response [24]; these patients were defined *arousable*. Another author, blinded to the arousal events during anesthesia, assessed the Brice modified questionnaire [25] for explicit recall 15 minutes after return of consciousness (in the Post-Anesthesia Care unit) and on the first post-operative day, in both cases in the quietest and most comfortable condition as possible.

## 2.3 Statistical analysis

We based the sample size calculation on the difference in CeP at ROC. Therefore, we used a pilot sample of 10 patients, five younger (age 18–64 years) and five older (age $\geq$65 years). We used following parameters: 0.2 µg ml$^{-1}$ difference in median CeP ROC between younger and older groups; 0.25 µg ml$^{-1}$ SD; a type I error probability of 0.05 and a type II error probability of 0.2. At least 25 patients were required for each group. We recruited patients consecutively until the necessary number in both groups was reached.

We used the Shapiro-Wilk test to test for normality. We report continuous, normally distributed variables as mean ± SD and continuous non-normally distributed variables as median [IQR] or minimum to maximum. We tested for differences between groups using the two-tailed Student t-test or two-tailed Mann-Whitney U test for normally and non-normally distributed variables, respectively. We report categorical variables as number (percentage), and tested for differences between groups using the Chi-square test. We supplement the findings from the t-test/Mann-Whitney U test with the area under the curve as effect size, together with 10k-fold bootstrapped 95% confidence intervals (CI). Therefore, we used the MATLAB-based MES toolbox [26].

We calculated a linear model to assess the relationships between age and selected variables with the MATLAB *fitlm* function. The strength and direction of association between two continuous variables were determined using the Pearson's correlation.

We calculated the Wilcoxon signed rank test to evaluate possible differences between CeP/CeR at ROC and arousal.

We used R version 3.4.0 (2017-04-21) and MATLAB R2017b (The Mathworks, Natick, MA) for statistical analyses. We considered $P < 0.05$ statistically significant.

## Results

### 3.1 Demographics

During the study period, 81 of 89 patients undergoing breast surgery provided written informed consent to join the study. Of these, 5 patients were excluded because of not meeting inclusion criteria, and 1 patient was excluded due to incomplete data. Data from 75 patients

were therefore analyzed. Their median age, weight, height, BMI and schooling years were 60 years (IQR:50–70), 65 kg (IQR:57–78), 161 cm (IQR:158–168), 25.1 kg m$^{-2}$ (IQR:20,6–29,6) and 13 years (IQR:8–13) respectively; ASA I/II was 37/38. Median surgery duration was 68.6 min (IQR:40–90). Table 1 contains the patient characteristics for the young and old patients. Older patients had significantly fewer schooling years and a higher number of previous general anesthesia.

## 3.2 Age effects on CeP and CeR

During anesthesia maintenance, we did not find any age-related difference or linear trend in median CeP or CeR concentrations: CeP was 3[IQR:2–3]μg ml$^{-1}$ in the old vs 2.5[IQR:2–3]μg ml$^{-1}$ in the young (p = 0,332) and CeR was 2[IQR:1.5–2.2]ng ml$^{-1}$ in the old vs 1.8[IQR:1.5–2.0] ng ml$^{-1}$ in the young (p = 0,384). The fentanyl doses did not differ significantly between the age groups. Fig 1A and 1B contain the linear model plots and the young vs old box plots. Table 1 contains the statistical details of the young versus old comparisons and Table 2 contains the details of the linear models.

At ROC, we found a statistically significant influence on age on CeP: *CeP = -0.01*age+1.2* (p<0.001) (old vs. young: 0.5[IQR:0.4–0.6]μg ml$^{-1}$ vs. 0.7[IQR:0.5–0.9]μg ml$^{-1}$ p = 0.010). We did not find a statistical difference between old and young for CeR (old vs. young: 0,3 ng ml$^{-}$

**Table 1. Patient, drug, and monitoring parameter characteristics according to age.**

| Variable | Age (years) | | P value | AUC value |
|---|---|---|---|---|
| | <65 (n = 50) | ≥65 (n = 25) | | |
| *Patient characteristics* | | | | |
| BMI (kg m$^{-2}$) | 25.1 [21.6 to 28.3], 16.8 to 35.3 | 25.6 [21.9 to 28], 17.4 to 32.8 | 0.996 | 1 [0.36–0.34] |
| ASA I/II (n) | 28/22 (56%/44%) | 9/16 (36%/84%) | 0.102* | N/A |
| Schooling (years) | 13 [8 to 13], 5 to 23 | 5 [5 to 8], 5 to 18 | <0.001 | 0.83 [0.72–0.93] |
| Previous GA (n) | 1 [0.25 to 2], 0 to 5 | 3 [1 to 4], 0 to 7 | 0.001 | 0.27 [0.16–0.40]] |
| Duration of surgery (min) | 60 [50 to 80], 20 to 180 | 45 [35 to 90], 25 to 200 | 0.154 | 0.60 [0.44–0.75] |
| Fentanyl (μg kg$^{-1}$ min$^{-1}$) | 0.04 [0.02 to 0.06], 0.01 to 0.17 | 0.03 [0.02 to 0.07], 0.01 to 0.11 | 0.562 | 0.54 [0.39–0.69] |
| Arousal reaction Y/N (n) | 13/37 (26%/74%) | 5/20 (20%/80%) | 0.566* | N/A |
| *Anesthesia maintenance* | | | | |
| CeP (μg ml$^{-1}$) | 2.5 [2 to 3], 1.5 to 4.5 | 3 [2 to 3], 1.5 to 4.5 | 0.332 | 0.43 [0.29–0.58] |
| CeR (ng ml$^{-1}$) | 1.8 [1.5 to 2.0], 0.6 to 2.8 | 2.0 [1.5 to 2.2], 1 to 3 | 0.384 | 0.43 [0.28–0.59] |
| RE | 47.5 [42 to 52.5], 22 to 85 | 55 [49.3 to 61], 33 to 95 | 0.003 | 0.29 [0.16–0.42] |
| SE | 45 [40 to 50], 20 to 75 | 50 [45 to 55,5], 30 to 88 | 0.005 | 0.30 [0.18–0.43] |
| RE–SE | 3 [2 to 5], 1 to 10 | 5 [2.8 to 5.5], 0 to 10 | 0.020 | 0.33 [0.21–0.49] |
| *ROC* | | | | |
| CeP (μg ml$^{-1}$) | 0.7 [0.5 to 0.9], 0.1 to 1.3 | 0.5 [0.4 to 0.6], 0.2 to 0.9 | 0.010 | 0.68 [0.56–0.80] |
| CeR (ng ml$^{-1}$) | 0.3 [0.2 to 0.5], 0.1 to 1.5 | 0.4 [0.2 to 0.5], 0 to 0.9 | 0.820 | 0.52 [0.36–0.67] |
| RE | 92.5 [86 to 95.25], 28 to 100 | 87 [67 to 95], 34 to 99 | 0.153 | 0.60 [0.46–0.74] |
| SE | 81.5 [71 to 86], 27 to 91 | 78 [61.5 to 85], 31 to 89 | 0.223 | 0.59 [0.45–0.72] |
| RE–SE | 10 [8 to 15], 0 to 36 | 10 [7.5 to 12.5], 0 to 27 | 0.424 | 0.56 [0.42–0.69] |

*Notes.* Data are median [IQR], minimum to maximum unless otherwise indicated. Arousal reaction was defined as 'light anesthesia' associated with involuntary movement, inadequate ventilation because of vocal cord closure, or a definite hemodynamic response (Sanders RD 2012).

*Chi-square p-value.

*Abbreviations.* BMI: Body mass index; CeP: Effective estimated brain concentration of propofol; CeR: Effective estimated brain concentration of remifentanil; GA: General anesthesia; RE: Response entropy (range, 0–100); SE: State entropy (range, 0–91); RE–SE: Difference between RE and SE; ROC: At return of consciousness.

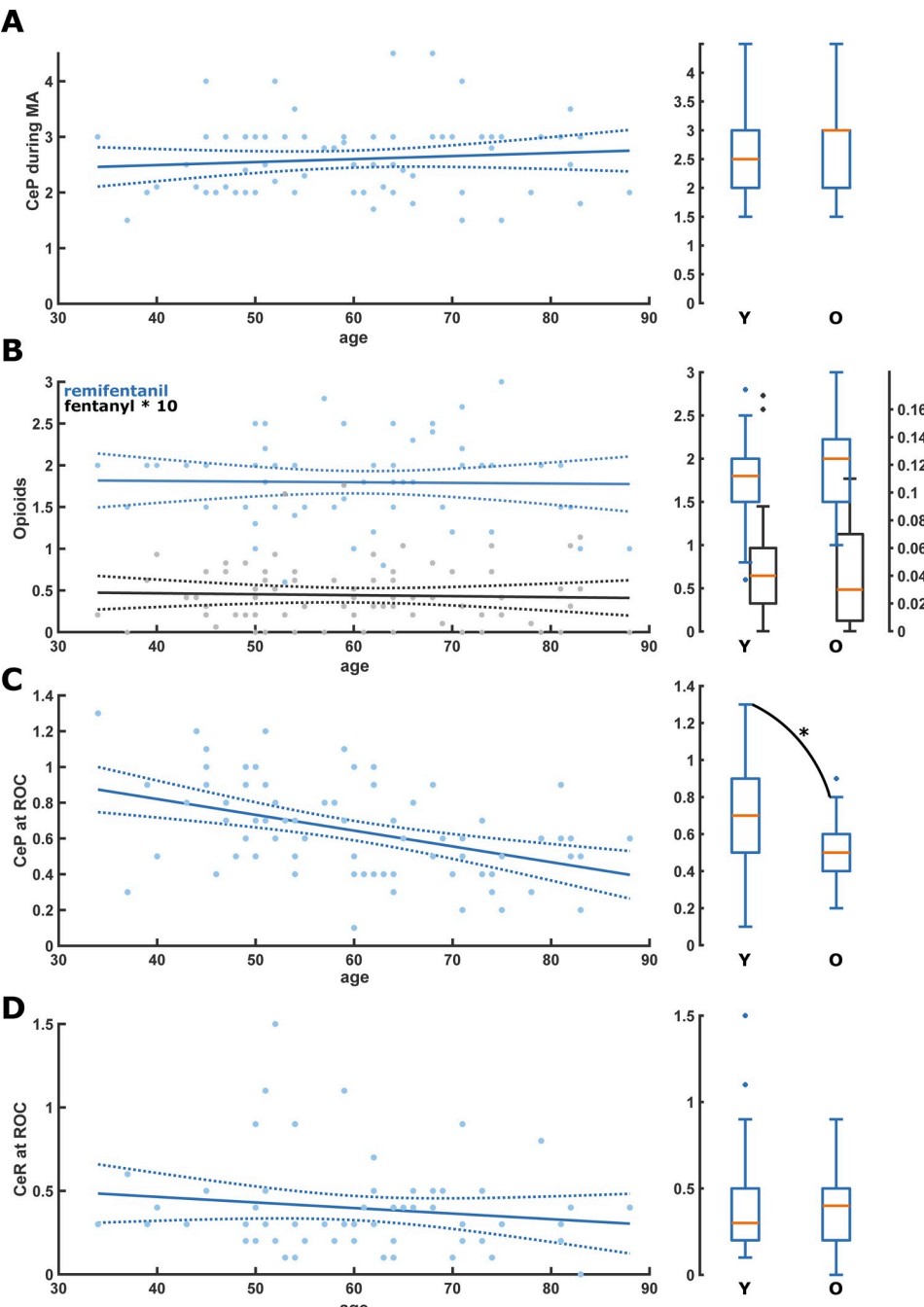

**Fig 1. Scatter plots with linear regression as well as box plots for the comparison of the young (Y: <65 yr) and old (O: ≥65 yr) for the different drugs during anesthesia maintenance and at return of consciousness (ROC).** A. There was no significant change or difference of CeP during anesthesia maintenance. B. There was no significant change or difference of CeR (blue) and fentanyl (black) during anesthesia maintenance. C. CeP at ROC was significantly lower in the old patients (p = 0.010; Area Under the Curve (AUC): 0.68 [0.56 0.80]); D. There was no significant change or difference of CeR at ROC. CeP: Effective estimated brain concentration of Propofol (μg ml$^{-1}$); CeR: Effective estimated brain concentration of Remifentanil (ng ml$^{-1}$); MA: Maintenance of anesthesia; ROC: Return of consciousness.

[1][IQR:0,2–0,5] vs 0,5 ng ml$^{-1}$[IQR:0.2–0.5], p = 0.820). Fig 1C and 1D and Tables 1 and 2 present the details.

**Table 2. Statistical parameters of the linear model.**

| Linear equation | t-statistic | P-value | Corr. coeff |
|---|---|---|---|
| *Anesthesia maintenance* | 0.88 | 0.382 | 0.10 |
| CeP = 0.005*age+2.277 | -0.14 | 0.887 | -0.02 |
| CeR = -0.001*age+1.846 | -0.33 | 0.741 | -0.04 |
| fent = -0.001*age+0.494 | 3.61 | <0.001 | 0.39 |
| RE = 0.364*age+28.217 | 3.46 | <0.001 | 0.38 |
| SE = 0.313*age+27.389 | 2.58 | 0.012 | 0.29 |
| RE-SE = 0.051*age+0.828 | -1.35 | 0.181 | -0.16 |
| SPI = -0.140*age+42.066 | 2.80 | 0.007 | 0.31 |
| max(RE-SE) = 0.116*max(SPI)+1.969 | | | |
| *ROC* | -4.04 | <0.001 | -0.43 |
| CeP = -0.009*age+1.174 | -1.12 | 0.268 | -0.15 |
| CeR = -0.003*age+0.598 | -0.40 | 0.694 | -0.05 |
| RE = -0.057*age+77.221 | -0.84 | 0.406 | -0.10 |
| SE = -0.126*age+93.119 | -1.11 | 0.272 | -0.13 |
| RE-SE = -0.070*age+15.898 | 0.88 | 0.382 | 0.10 |

*Abbreviations*: CeP: Effective estimated brain concentration of propofol; CeR: Effective estimated brain concentration of remifentanil; ROC: Return of consciousness.

## 3.3 Age effects on processed EEG

When looking at the Entropy values during maintenance, we found an age-related increase in mean RE and SE: *RE = 0.36*age+28.22* (p<0.001); *SE = 0.31*age+27.39* (p<0.001). Consequently, RE and SE were significantly higher in the old: RE: 55[IQR:49.3–61] vs 47.5[IQR:42–52.5], p = 0.003; SE: 50[IQR:45–55.5] vs 45[IQR:40–50], p = 0.005). The RE-SE also increased with age, *RE-SE = 0.05*age+0.83* (p = 0.012); Old vs. young: Median RE-SE (5[IQR:2.8–5.5] vs 3[2–5], p = 0.020). Fig 2A and 2B and **Tables 1 and 2** contain the detailed information. We did not observe significant differences in the entropies with age at ROC (Fig 2C and 2D and **Tables 1 and 2**). Further, there was no significant age effect on the mean SPI during maintenance (S2A Fig).

## 3.4 PK/PD and processed EEG differences between arousers and non-arousers

No patients exhibited explicit recall 15 minutes after ROC or at the first post-operative day after the interview with the Brice questionnaire. An arousal reaction maintenance occurred in 18 of 75 patients (24%), with no significant difference (p = 0.566) in occurrence rate between the young and old patient group. We did not observe a significant difference in CeP and CeR in patients with and without arousal reaction (Fig 3A and 3B). Median CeP during mainte-nance of the arousable patients was 2.45 µg ml⁻¹[IQR:2–3µg ml⁻¹] and it was 2 µg ml⁻¹[IQR:1.7–2.8µg ml⁻¹] at the arousal reaction (p = 0.069). This was not statistically different from the median CeP during maintenance of the non-arousable patients with 2.5 µg ml⁻¹[IQR:2–3µg ml⁻¹], p = 0.596. We found a significant difference (p<0.001) between CeP at the arousal reaction (2 µg ml⁻¹[IQR:1.7–2.8µg ml⁻¹]) and at ROC in the arousable patients (0.67 µg ml⁻¹[IQR:0.4–0.9 0.67 µg ml⁻¹]). Median CeR during maintenance in the arousable patients (1.9 ng ml⁻¹[IQR:1.5-2ng ml⁻¹]), as well as their CeR at the arousal reaction (2 ng ml⁻¹[IQR:1.5-2ng ml⁻¹]) was not statistically different if compared with median CeR during main-tenance of the non-arousable patients (2 ng ml⁻¹[IQR:1.1–2.4ng ml⁻¹], p = 0.842 and p = 0.603, respectively) as presented in Fig 3B and Table 3. In addition, median CeR at arousal reaction

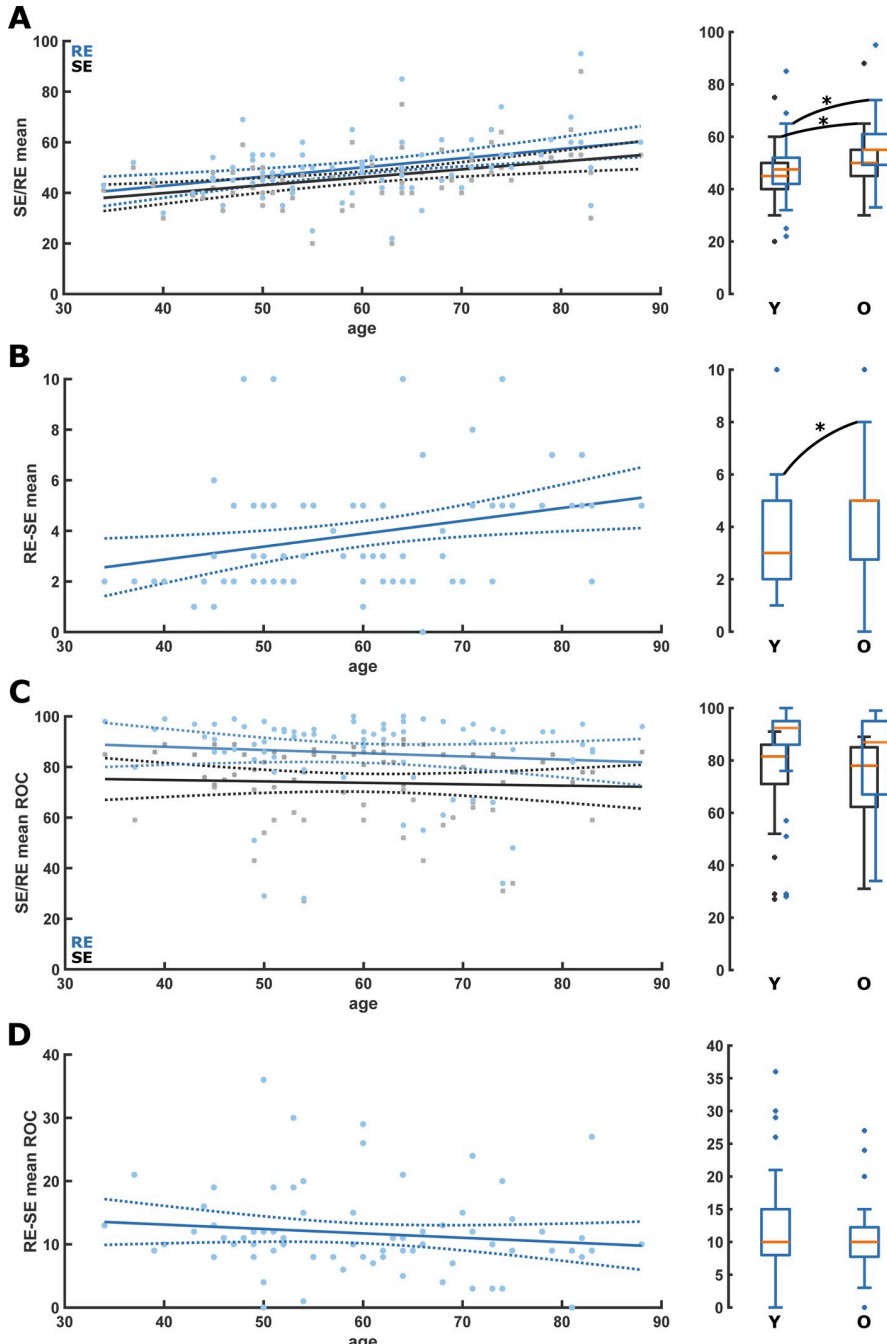

**Fig 2. Scatter plots with linear regression as well as box plots for the comparison of the young (Y: <65 yr) and old (O: ≥65 yr) for the state (SE) and response Entropy (RE) during anesthesia maintenance and at return of consciousness (ROC).** A. RE (blue) and SE (black) were significantly higher in the old during anesthesia maintenance. SE: p = 0.005; AUC = 0.30 [0.18 0.43]; RE: p = 0.003; AUC = 0.29 [0.16 0.42]; B. RE-SE was significantly (p = 0.016; AUC = 0.33 [0.21 0.47]) higher in the old during anesthesia maintenance. C. There was no significant change or difference of RE (blue) and SE (black) at ROC. D. There was no significant change or difference of RE-SE at ROC. RE: Response Entropy; SE: State Entropy; MA: Maintenance of anesthesia; ROC: Return of consciousness.

was significantly different if compared with the CeR at ROC in the same arousable patients (2 ng ml$^{-1}$[IQR:1.1–2.4 ng ml$^{-1}$] vs 0.4 μg ml$^{-1}$[IQR:0.3–0.5 ng ml$^{-1}$], p = 0.005). Regarding the

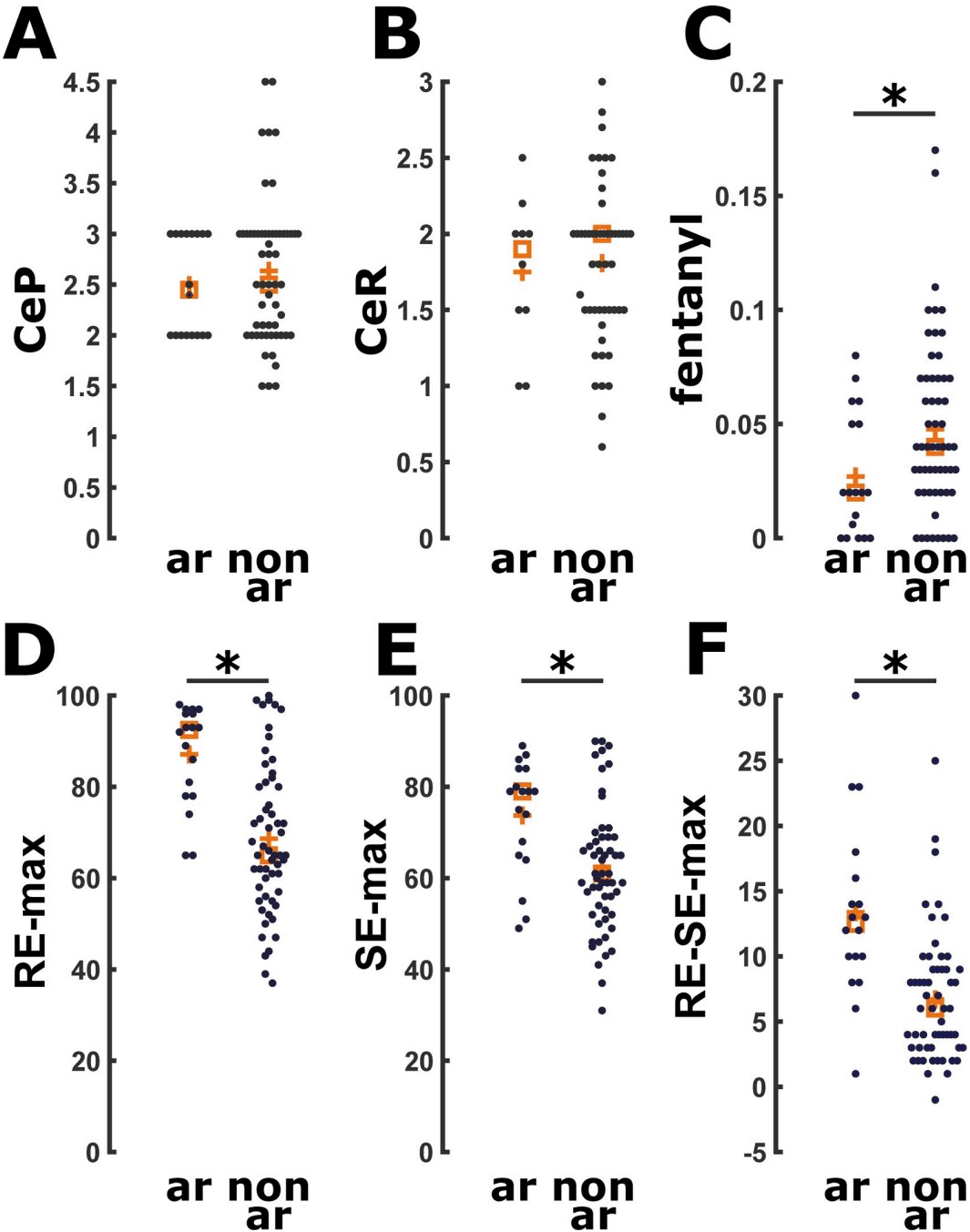

**Fig 3. Scatter plots for drug and state (SE) and response (RE) values comparing arousers (ar) versus non-arousers (n-ar).**
A. There was no significant difference in CeP (µg ml$^{-1}$) during anesthesia maintenance. B. There was no significant difference in CeR (ng ml$^{-1}$) during anesthesia maintenance. C. Arousers received significantly less fentanyl (mg/kg/min) (p = 0.026; AUC = 0.33 [0.19 0.47]). D. Arousers had significantly higher RE (p<0.001; AUC = 0.80 [0.70 0.90]). E. Arousers had significantly higher SE (p = 0.003; AUC = 0.73 [0.59 0.86]). F. Arousers had significantly higher RE-SE (<0.001; AUC = 0.81 [0.68 0.92]). CeP: Effective estimated brain concentration of Propofol (µg ml$^{-1}$) during maintenance of anesthesia; CeR: Effective estimated brain concentration of Remifentanil (ng ml$^{-1}$) during maintenance of anesthesia; ar: Arousers; n-ar: Non-arousers.

fentanyl dose, arousable patients received a significantly (p = 0.027) lower dosage of fentanyl 0.02 µg kg$^{-1}$min$^{-1}$[IQR:0.02–0.05µg kg$^{-1}$ min$^{-1}$] than non-arousable patients (0.04 µg kg$^{-1}$min$^{-}$

**Table 3. Characteristics of patients according to presence of an arousal reaction.**

| Variable | Arousal reaction | | P value | AUC value |
|---|---|---|---|---|
| | No (n = 57) | Yes (n = 18) | | |
| *Patient characteristics* | | | | |
| Age (years) | 61 [52 to 69], 34 to 83 | 55.5 [49 to 70.8], 39 to 88 | 0.572 | 0.46 [0.29–0.63] |
| BMI (kg m$^{-2}$) | 24.2 [21.6 to 27.5], 16.8 to 34.5 | 27.1 [24.4 to 29.6], 17.4 to 35.3 | 0.130 | 0.62 [0.45–0.77] |
| ASA I/II | 30/27 (52.6%/47.4%) | 7/11 (38.9%/61.1%) | 0.310* | N/A |
| Years of schooling (years) | 13 [8 to 13], 5 to 23 | 8 [8 to 13], 5 to 18 | 0.675 | 0.47 [0.32–0.62] |
| Previous GA (n) | 1 [1 to 3], 0 to 7 | 2 [1 to 2.75], 0 to 5 | 0.689 | 0.53 [0.39–0.67] |
| Duration of surgery (min) | 60 [45 to 80], 20 to 200 | 62.5 [41.2 to 103.7], 35 to 180 | 0.426 | 0.56 [0.41–0.72] |
| Fentanyl (µg kg$^{-1}$ min$^{-1}$) | 0.04 [0.02 to 0.07], 0.01 to 0.17 | 0.02 [0.02 to 0.05], 0.01 to 0.08 | 0.027 | 0.33 [0.19–0.47] |
| *Anesthesia maintenance* | | | | |
| CeP (µg ml$^{-1}$) | 2.5 [2 to 3], 1.50 to 4.5 | 2.45 [2 to 3], 2 to 3 | 0.596 | 0.46 [0.32–0.60] |
| CeR (ng ml$^{-1}$) | 2.0 [1.5 to 2.0], 0.6 to 3.0 | 1.9 [1.5 to 2], 1 to 2.5 | 0.842 | 0.48 [0.29–0.67] |
| RE | 48 [42 to 55], 22 to 95 | 51.5 [47,3 to 56,3], 35 to 85 | 0.126 | 0.62 [0.48–0.75] |
| SE | 45 [40 to 50], 20 to 88 | 49 [44.5 to 51.3], 33 to 75 | 0.091 | 0.63 [0.50–0.76] |
| RE–SE | 3.9 [2 to 5], 1 to 10 | 4 [2 to 5], 1 to 10 | 0.409 | 0.49 [0.34–0.64] |
| RE max | 65 [56.5 to 80.5], 37 to 100 | 92.5 [78 to 96.3], 65 to 98 | <0.001 | 0.80 [0.70–0.90] |
| SE max | 61.0 [52.5 to 69], 31 to 90 | 79 [64.8 to 84], 51 to 89 | 0.003 | 0.73 [0.59–0.86] |
| RE–SE max | 6 [3 to 9], 1 to 25 | 12.5 [9.5 to 16.5], 1 to 30 | <0.001 | 0.81 [0.68–0.92] |
| SPI | 30 [25 to 40], 15 to 60 | 37.5 [30 to 44.2], 17 to 60 | 0.159 | 0.61 [0.46–0.75] |
| SPI max | 52.5 [40 to 66.2], 24 to 83 | 63.5 [49.5 to 74.7], 42 to 89 | 0.030 | 0.67 [0.53–0.80] |

*Notes.* Data are median (IQR), minimum to maximum unless otherwise indicated. Arousal reaction was defined as "light anesthesia" associated with involuntary movement, inadequate ventilation because of vocal cord closure, or a definite hemodynamic response (Sanders RD 2012).

*Chi-squared p-values.

*Abbreviations.* BMI: Body mass index; CeP: Effective estimated brain concentration of propofol; CeR: Effective estimated brain concentration of remifentanil; GA: General anesthetics; max: Maximum; RE: Response entropy (range, 0–100); SE: State entropy (range, 0–91); RE–SE: Difference between RE and SE; SPI: Surgical Pleth Index (range, 0–100).

[1][IQR: 0.02–0.07µg kg$^{-1}$ min$^{-1}$]) (Fig 3C). When analyzing the pEEG indices, we found higher maximum (but not mean) RE (p<0.001), SE (p = 0.003), RE–SE (p<0.001; 0.81[0.68–0.92]) during maintenance in the arousable patients (Fig 3D–3F and Table 3). We also found higher maximum, but not mean, SPI values during maintenance (p = 0.030) in the arousable patients. (S3 Fig and Table 3). There was a significant positive (p = 0.006) correlation between maximum SPI and maximum RE–SE difference during surgery, i.e., max(*RE-SE*) = 0.116*max(*SPI*) +1.969. S2B Fig displays this relationship.

## Discussion

Our results confirm previous findings about age-influence on CeP at ROC: older patients regain consciousness at lower CeP [6,7]. We couldn't find significant differences in the median CeP during Entropy-guided TIVA-TCI maintenance described by previous trials [8], but meanwhile we observed an age-induced change in the pEEG parameters: RE, SE, and RE-SE were significantly higher during maintenance in the elderly. Further, we couldn't find age-related differences in CeP at arousal reactions. Arousable patients expressed significantly higher maximum RE–SE difference values.

Hence, we could reveal an age-influence of the pEEG-guidance on anesthesia navigation. Without targeting the anesthetic level to the recommended index range, CeP of our older

patients might have been lower [3]. In our study, targeting the level to a SE of 40–60, it wasn't. This is the case, because age-induced changes in the EEG seem to influence the pEEG indices.

## 4.1 Characteristic of patients according to age

Pharmacodynamics and pharmacokinetics of anesthetics are affected by age [2–6,16], and an increased Propofol-sensitivity in older patients may explain the lower CeP at ROC observed in our and previous studies [6,7]. However, we could not find the expected age-related difference in CeP during maintenance, probably because of the use of SE to target the anesthetic level. EEG amplitude under general anesthesia decreases with age and composition of recorded EEG changes towards a stronger contribution of higher frequencies [18,19]. EEG architecture's changes could explain our higher Entropy values observation in elders during anesthesia maintenance, even in the case of actively targeting the range. Higher indices with age during maintenance were described for the Bispectral Index (BIS) and for the spectral Entropy [18], the underlying method for SE and RE calculation [9]. Here, we can confirm the impact of age for SE and RE. Further, the increase in RE-SE with age during maintenance is indicative of a stronger, age-related, contribution of higher frequencies in the 32 Hz to 47 Hz range processed by RE [9].

Age influence on SE and RE probably is the reason for our older patients having a higher CeP during maintenance, because anesthesiologists navigated anesthesia by the index. Hence, our results highlight a possible conflict between TCI and EEG-based indices for elders. Consequently, more hypnotics than required were possibly given to older patients, increasing the risk of hypnotics overdosage in this population. The case of one 82-year-old patient with an adequate CeP during maintenance (2.5 μg ml$^{-1}$) showing median RE and SE of 95 and 88 during maintenance most probably highlights another issue: undetected EEG burst suppression (BSupp). BSupp indicate severe brain neuronal activity and metabolic rate reduction that may increase postoperative delirium and POCD risk [27,28]. pEEG indices seem to have limited ability to reliably detect BSupp patterns, underestimating their occurrence [29]. If undetected, the suppressed EEG low-amplitude and high frequency characteristics can cause high index values [30]. This may have been our patient's case since she showed no arousal reaction during surgery and had no awareness with explicit recall. In order to generally identify BSupp, even if the indices do not, a visual identification of the raw EEG or its spectral representation was suggested [29]. This case also highlights that Propofol concentrations that are usually appropriate can cause persisting BSupp in certain cases. Hence, EEG-based patient monitoring can help to identify these cases and to individually adjust the anesthetic level. But, using processed EEG indices only to guide TCI anesthesia with Propofol may expose older patients to higher than necessary doses that could lead to a higher risk of adverse outcomes such as postoperative neurocognitive disorders [10].

## 4.2 Characteristic of patients according to arousal reaction

Response surface models revealed prominent synergistic effects between Propofol and Remifentanil in blunting the response to nociceptive stimuli and to avoid arousal reactions [21]. In our study, *arousable* patients had adequate CeP and CeR values during maintenance [1]. However, they received less fentanyl during surgery, and they had a higher median SPI during maintenance. This suggests that, even within the suggested TCI interval [23], analgesic management may have been inadequate, and SPI threshold of 30 suggested for inhalational anesthesia [21] should be adopted also for TCI anesthesia in order to prevent an arousal reaction's risk.

Entropy monitoring may reduce the likelihood of a reaction to intraoperative nociceptive stimulus [31]. In our study, *arousable* patients had higher maximum RE-SE RE, SE, and SPI values than *non-arousable* patients, suggesting monitoring devices' detection of arousal

reaction. Because we didn't see differences in median values between arousable and non-arousable patients, the predictive power for a possible arousal may be low.

Painful stimulation increases cardiovascular activity, detected by SPI [21]. However, muscle tension increases, another facet of the response to painful stimulation [32], also occur because of proprioceptive inputs to the reticulo-thalamic activating system [33]. SE, computed over the EEG-dominant frequency range (0.8–32 Hz), mainly reflects the patient's cortical state. By contrast, RE is computed over the EEG- and EMG-dominant frequency ranges (0.8–47 Hz) and thereby at least partly serves as an indicator of upper facial EMG activation, which has been reported to represent nociception or impending awakening [31]. Increment in RE–SE, even if it occurs only transiently because of a subsequent increase in SE, may be useful for identifying an inadequate anesthetic/analgesic plan with arousal reaction's risk following painful stimulation. So in case of increasing EMG the SE and RE will start to drift apart because of an increased influence of high frequency oscillatory activity picked up by RE, but not SE. Another case, more important to our investigations is the age-related flattening of the Power spectral density under general anesthesia [18]. This flattening also leads to an increased influence of higher frequencies that will also increase the difference between the RE and SE. So we are confident that the observed increase in the RE-SE with age under general anesthesia is not caused by increase EMG with age, but by the age-induced increased of higher EEG frequencies. Hence, the indices seem to appropriately respond to an arousal reaction after noxious stimulation independent of age.

### 4.3 Limitations

This study has some limitations. Firstly, it was conducted only on female patients undergoing breast surgery: future trials should confirm our observations also on males and other surgery-types; in addition, although all of our patients were in the menopausal period, we did not account for hormonal contribution in our analysis. Secondly, SE and RE trends, and the raw EEG tracings, weren't electronically stored, so we couldn't analyze RE and SE values during arousal reaction: we can only speculate that the observed maximum RE and SE in arousers were its consequence. Analysis of raw EEG data could have provided additional insights into the arousal reaction's nature. Raw EEG data could have provided additional informationa also on BSupp, that can alter SE and RE values, however because the vast majority of our patients had indices in the index interval reflecting "adequate anesthesia", i.e., between 40 and 60, we are very confident that BSupp is not an issue in our data.

### Conclusion

Estimated concentrations of Propofol during anesthesia maintenance seem not to be age-related in patients during TIVA-TCI for breast surgery when anesthesia is navigated by the processed EEG index, highlighting the impact on age on general anesthesia. The SE and RE seem to increase with age during adequate levels of anesthesia.

The occurrence of an arousal reaction seems independent of age, but lower CeR during anesthesia maintenance might lead to an increased risk of an arousal reaction. Finally, additional information as for instance provided by the SPI could help to improve intraoperative patient monitoring and to optimize the hypnotic and analgesic component of anesthesia.

### Supporting information

**S1 Fig. CONSORT flow diagram showing selection of the enrolled patients.**
(TIF)

**S2 Fig.** Scatter plots with linear regression for the A) mean SPI versus age (years) and B) max (SPI) versus max(RE-SE) relationship during anesthesia maintenance. There was no significant change of the mean SPI with age. There was a significant increase of the max (RE-SE) with increase of the maximum SPI observed during anesthesia maintenance. SPI: Surgical Pleth Index; RE: Response entropy; SE: State Entropy.
(TIF)

**S3 Fig. Scatter plots for mean and maximum SPI values comparing arousers (ar) versus non-arousers (n-ar).** There was no significant difference of mean SPI between arousers and non-arousers during anesthesia maintenance. The maximum SPI was significantly higher in arouser during anesthesia maintenance (p = 0.03; AUC = 0.67 [0.53–0.80]). SPI: Surgical Pleth Index; ar: arousers; n-ar: non-arousers.
(TIF)

**S1 Database.**
(TIF)

## Acknowledgments

Authors would like to thank Paolo Burelli, MD, Chief of the Breast Unit of Treviso Regional Hospital, and Domenico Gentile, MD, Alessandro de Laurenzis, PsyD, Lisa Entilli, MD and Cristina Gioia, MD of the Department of Anesthesiology and Intensive Care of Treviso Regional Hospital for their assistance with the study.

## Author Contributions

**Conceptualization:** Federico Linassi, Eleonora Maran, Paolo Zanatta.

**Data curation:** Federico Linassi, Matthias Kreuzer, Eleonora Maran, Antonio Farnia, Paolo Navalesi, Michele Carron.

**Formal analysis:** Federico Linassi, Matthias Kreuzer, Michele Carron.

**Funding acquisition:** Michele Carron.

**Investigation:** Federico Linassi, Eleonora Maran, Antonio Farnia, Michele Carron.

**Methodology:** Federico Linassi, Eleonora Maran, Antonio Farnia, Paolo Navalesi, Michele Carron.

**Project administration:** Federico Linassi, Antonio Farnia, Paolo Zanatta, Paolo Navalesi, Michele Carron.

**Resources:** Federico Linassi, Paolo Zanatta, Paolo Navalesi, Michele Carron.

**Software:** Federico Linassi, Matthias Kreuzer.

**Supervision:** Federico Linassi, Matthias Kreuzer, Antonio Farnia, Paolo Navalesi, Michele Carron.

**Validation:** Federico Linassi, Matthias Kreuzer, Paolo Zanatta, Michele Carron.

**Visualization:** Federico Linassi, Paolo Zanatta, Paolo Navalesi.

**Writing – original draft:** Federico Linassi, Matthias Kreuzer, Michele Carron.

**Writing – review & editing:** Matthias Kreuzer, Eleonora Maran, Antonio Farnia, Paolo Zanatta, Michele Carron.

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
