## [Decision Letter · Decision Letter 0]

8 Sep 2020

PONE-D-20-16382

Age influences on Propofol estimated brain concentration and Entropy during maintenance and at return of consciousness during Total Intravenous Anesthesia With Target-Controlled Infusion in Unparalyzed Patients: an Observational Prospective Trial

PLOS ONE

Dear Dr. Linassi,

Thank you for submitting your manuscript to PLOS ONE. After careful consideration, we feel that it has merit but does not fully meet PLOS ONE’s publication criteria as it currently stands. Therefore, we invite you to submit a revised version of the manuscript that addresses the points raised during the review process.

The authors described an observational prospective trial to validate the aging effects on effective estimated brain concentration of propofol during TIVA-TCI Entropy-guided anesthesia. The manuscript has been assessed by three reviewers and their comments are available below. The reviewers both concerned more details regarding the results.

We look forward to receiving your revised manuscript.

Kind regards,

JianJun Yang, M.D., Ph.D.

Academic Editor

PLOS ONE

Journal Requirements:

2. "Please explain why the CT registration (NCT04129112) status is still 'recruiting

5. Please include your tables as part of your main manuscript and remove the individual files. Please note that supplementary tables (should remain/ be uploaded) as separate "supporting information" files

Reviewers' comments:

Reviewer's Responses to Questions

**Comments to the Author**

1. Is the manuscript technically sound, and do the data support the conclusions?

Reviewer #1: Partly

Reviewer #2: Partly

Reviewer #3: Yes

2. Has the statistical analysis been performed appropriately and rigorously? 

Reviewer #1: I Don't Know

Reviewer #2: No

Reviewer #3: Yes

3. Have the authors made all data underlying the findings in their manuscript fully available?

Reviewer #1: Yes

Reviewer #2: Yes

Reviewer #3: Yes

4. Is the manuscript presented in an intelligible fashion and written in standard English?

Reviewer #1: Yes

Reviewer #2: Yes

Reviewer #3: Yes

5. Review Comments to the Author

Reviewer #1: In this manuscript, the authors examined the effect of aging on the pharmacodynamics/pharmacokinetics of propofol and remifentanil, in particular as they relate to

Entropy-guided total intravenous anesthesia with target-controlled infusions (TIVA-TCI) They compared aging effects on effective estimated brain concentration of Propofol (CeP) during TIVA-TCI Entropy-guided anesthesia, without neuromuscular blockade (NMB). Secondary endpoints included the relationships between age and CeP at arousal reaction (AR), return of consciousness (ROC) and explicit recall. They calculated a linear model to evaluate the impact of age on observational variables and performed pairwise tests to compare older (≥65 years, n=50) and younger (<65 years, n=25) patients as well as patients with and without an an arousal response. Their results suggest that there is no age difference in CeP during anesthesia maintenance, but the CeP significantly decreased with age for return of consciousness. Interestingly, entropy values during MA increased with age despite comparable CeP's. The authors are clearly experienced and well versed in the field, but their work seemingly contradicts their previous work as well as that of others regarding increased sensitivities of the elderly to anesthetics, especially during TIVA. As such, there are several questions that need to be addressed:

1. The authors only enrolled women in their cohort. Since we know that MAC is altered by hormonal contributions, most notably progesterone in the pregnant patient, did they account for any hormonal contributions related to menstrual cycle and/or menopause. If not, could they at least comment on this?

2. It makes sense that older age, with an implied increased sensitivity to anesthetics, would demonstrate a lower CeP at ROC, but why is this same sensitivity not found in higher doses required for true anesthesia in contradiction to their previous work and that of others?

3. Similarly, if older age is associated with increased sensitivity to anesthetics, why would the entropy values for wakefulness be higher in the elderly at comparable CeP? Doesn't this contradict the ROC data?

4. We have had significant experience with the BIS and entropy measures of consciousness and have always found a degree of EMG contamination, which the static entropy is "supposed" to filter out. Can the authors comment on this given that the patients were not paralyzed?

5. How does this study differ from those mentioned in the following and why are the results so different:"CeP at loss of consciousness (LOC) and return of consciousness (ROC) during TIVA-TCI are lower in the elderly (age ≥65 years) than in youngers (age <65 years).[6,7] Further, it seems that lower CeP is necessary to maintain a similar anesthetic level in elderly using TIVA-TCI during cardiac surgery,[8] suggesting that older patients are more sensitive to Propofol administered for anesthesia maintenance."

6. While formal PK/PD models for propofol and remifentanil have been established for quite some time, it is also well known that the variability in seeminly adequate effect site concentrations is quite high. Can the authors comment on this variability in their population?

7. Can the authors comment more on their fentanyl doses as a confounding factor? Differential dosing here could grossly effect their results if not properly controlled. For instance, is the lack of difference in CeP and CeR between old and young patients during maintance of anesthesia possibly due to the younger patients receiving more fentanyl? They already state that such is the case in arousable patients: "Regarding the fentanyl dose, arousable patients received a significantly (p=0.027,AUC:0.26[0.19-0.47]) lower dosage of fentanyl..."

8. As I have no expertise in the statistics used, a more formal review of such by a statistician is warranted, especially in light of the significant scatter noted in the figures.

9. I am also confused by the conclusion that there is do difference in the CeR and CeP during maintenance, yet the entropy is higher in the elderly. This makes the older patients sound like they were merely not as deeply anesthetized compared to the younger patients at similar predicted Ce's and would actually suggest a decreased sensitivity to the anesthetic in the elderly!

10. On page 10, without referring to the figures, it is not clear which groups are being compared in the statements "When analyzing the pEEG indices, we found higher maximum (but not mean) RE (p<0.001;AUC:0.80[0.70-0.90]), SE (p=0.003; AUC:0.73[0.59-0.86]), RE–SE (p<0.001; 0.81[0.68-0.92])during maintenance (Figure 4D-F; Table 3). We also found higher maximum, but not mean, SPI values during maintenance (p=0.030; AUC:0.67[0.46-0.75]) (Figure S2, Table 3). There was a significant positive (p=0.006) correlation between maximum SPI and maximum RE–SE difference during surgery, i.e., max(RE-SE)=0.116*max(SPI)+1.969."

11. The authors' concerns regarding burst suppression are important ones. As such, they must comment on the presence or absence of burst suppression in all of the patients as this could give the very misleading result to which they refer, and will completely alter their interpretation of the EEG comparisons. I understand that they may not have the raw EEG data stored, but if most of the elderly are in fact more sensitive to anesthetics, thereby inducing burst suppression, this could be responsible for a titration to an SE and RE that are contaminated. If in burst suppression, the elderly might have needed far less anesthetic than given thereby altering their conclusions.

small corrections:

page 5: than in youngers, "youngers" is not a word.

page 5: helping do avoid should be helping to avoid

page 7: number of previous general anesthesia should read anesthetics

page 7: schooling’s years should read years of schooling

page 9: previous general anesthesia. should read anesthetics

Reviewer #2: Some aspects of the statistical analyses had me a bit confused. I have made some comments below in an attempt to highlight some of the issues. Generally speaking, the analyses are pretty straightforward.

1. The abbreviation AUC is not defined anywhere in the manuscript. If this happens to be area under the ROC curve, this is not mentioned in the methods, I don't understand the usage in this manuscript, and I recommend cutting it unless you can provide justification. It also makes the text of the results section too number-heavy and I would consider removing from the text regardless.

2. The biggest problem I had with the analyses was the switching between assumptions for the outcome variables. Median and IQR summary statistics and Wilcoxon are used when the normality assumption is suspect. Linear regression and Pearson correlation coefficients are used when a normality assumption is permissible to make. For instance, in section 3.2, in the first paragraph, CeP and CeR are all reported as median IQR (with p-values from Mann-Whitney, if I understood correctly). This switches in the second paragraph to assuming normality with for the linear model, and then back to a nonparametric test for testing young v. old. Neither are necessarily wrong, but the consistency has me baffled. Maybe more explanation for these choices in the methods section would be helpful.

3. I also found it very confusing that both a linear model and a Pearson correlation were used. There are differences between the two (mainly that both variables in correlation are considered to be sampled with error, whereas only the outcome (Y) variable is in a linear model), but there is nothing in the methods to support the use of both. Personally, I would report the slopes of the linear regression models. Intercepts are sometimes not reported so I will defer to you as to whether that is needed.

4. Did you consider fitting multivariable regression models to account for any potential confounders?

5. In the figures, please ensure that the y-axis range for the scatter plot matches the y-axis range for the boxplots.

6. Also, I suggest expanding the y-axis to zero for all plots, e.g, Figure 2 A.

Reviewer #3: I am quite impressed with the methodological approach that you took, and the care that was displayed in its execution.

Using exacting methods you quantified an important domain of clinical practice and have brought illumination to a foggy domain that was sorely in need of scientific foundation.

That one of the investigators was involved in hands-on data acquisition was a bit troubling and I believe you need to demonstrate absence of potential bias in their data recording.

The statistical analysis was appropriate in all regards.

The data capture excellent and sample size more than ample.

My overall recommendation is to accept this fine work of science that has virtually immediate clinical relevance and application.

6. PLOS authors have the option to publish the peer review history of their article (what does this mean?). If published, this will include your full peer review and any attached files.

Reviewer #1: No

Reviewer #2: No

Reviewer #3: **Yes: **CJ Biddle

---

## [Author Response · Author response to Decision Letter 0]

27 Oct 2020

Padova, October 11th, 2020

Re: PONE-D-20-16382 R1

Title: Age influences on Propofol estimated brain concentration and Entropy during maintenance and at return of consciousness during Total Intravenous Anesthesia With Target-Controlled Infusion in Unparalyzed Patients: an Observational Prospective Trial

To:

Joerg Heber, M.D., Ph.D.

Editor in chief

PLOS ONE

and 

JianJun Yang, M.D., Ph.D.

Academic Editor 

Dear Prof Joerg Heber, M.D., Ph.D., and JianJun Yang, M.D., Ph.D.,

We are submitting a revised version of the above-mentioned manuscript. We wish to thank the Editor-in-chief and the Reviewers for the time spent to review our work and for the considerations and suggestions that have been included in the current version. We are happy to clarify previously unclear aspects of our manuscript in a point to point fashion.

Journal Requirements: 

Q1. Please ensure that your manuscript meets PLOS ONE's style requirements, including those for file naming. The PLOS ONE style templates can be found at

R1, We made changes as requested.

Q2. "Please explain why the CT registration (NCT04129112) status is still 'recruiting

R2. We are sorry for this, there was a delay on the update of CT. The status has been changed.

Q3. We note that you have indicated that data from this study are available upon request. PLOS only allows data to be available upon request if there are legal or ethical restrictions on sharing data publicly. For information on unacceptable data access restrictions, please see http://journals.plos.org/plosone/s/data-availability#loc-unacceptable-data-access-restrictions.

R3. We made changes as requested, we uploaded the anonymized data set.

Q4. PLOS requires an ORCID iD for the corresponding author in Editorial Manager on papers submitted after December 6th, 2016. Please ensure that you have an ORCID iD and that it is validated in Editorial Manager. To do this, go to ‘Update my Information’ (in the upper left-hand corner of the main menu), and click on the Fetch/Validate link next to the ORCID field. This will take you to the ORCID site and allow you to create a new iD or authenticate a pre-existing iD in Editorial Manager. Please see the following video for instructions on linking an ORCID iD to your Editorial Manager account: https://www.youtube.com/watch?v=_xcclfuvtxQ

R4. We made changes as requested.

Q5. Please include your tables as part of your main manuscript and remove the individual files. Please note that supplementary tables (should remain/ be uploaded) as separate "supporting information" files

R5. We made changes as requested.

Reviewers' comments: 

Reviewer's Responses to Questions

Comments to the Author

1. Is the manuscript technically sound, and do the data support the conclusions?

Reviewer #1: Partly

Reviewer #2: Partly

Reviewer #3: Yes

2. Has the statistical analysis been performed appropriately and rigorously?

Reviewer #1: I Don't Know

Reviewer #2: No

Reviewer #3: Yes

3. Have the authors made all data underlying the findings in their manuscript fully available?

Reviewer #1: Yes

Reviewer #2: Yes

Reviewer #3: Yes

4. Is the manuscript presented in an intelligible fashion and written in standard English?

Reviewer #1: Yes

Reviewer #2: Yes

Reviewer #3: Yes

5. Review Comments to the Author

Reviewer #1: 

In this manuscript, the authors examined the effect of aging on the pharmacodynamics/pharmacokinetics of propofol and remifentanil, in particular as they relate to

Entropy-guided total intravenous anesthesia with target-controlled infusions (TIVA-TCI) They compared aging effects on effective estimated brain concentration of Propofol (CeP) during TIVA-TCI Entropy-guided anesthesia, without neuromuscular blockade (NMB). Secondary endpoints included the relationships between age and CeP at arousal reaction (AR), return of consciousness (ROC) and explicit recall. They calculated a linear model to evaluate the impact of age on observational variables and performed pairwise tests to compare older (≥65 years, n=50) and younger (<65 years, n=25) patients as well as patients with and without an an arousal response. Their results suggest that there is no age difference in CeP during anesthesia maintenance, but the CeP significantly decreased with age for return of consciousness. Interestingly, entropy values during MA increased with age despite comparable CeP's. The authors are clearly experienced and well versed in the field, but their work seemingly contradicts their previous work as well as that of others regarding increased sensitivities of the elderly to anesthetics, especially during TIVA. As such, there are several questions that need to be addressed:

Q1. The authors only enrolled women in their cohort. Since we know that MAC is altered by hormonal contributions, most notably progesterone in the pregnant patient, did they account for any hormonal contributions related to menstrual cycle and/or menopause. If not, could they at least comment on this?

R1. We thank the reviewer for this comment. We did not take hormonal contributions into account for our analysis. We know that there are contrasting results in the literature not only on MAC but also on TIVA and estrogen levels (Basaran et al, 2019, Yavutz et al, 2007), and their role on TIVA-TCI anaesthesia is debated. However, all of the patients were in the menopause period so we postulate that there was not high heterogeneity in our sample from a hormonal point of view. However, we added this in the limits section.

Q2. It makes sense that older age, with an implied increased sensitivity to anesthetics, would demonstrate a lower CeP at ROC, but why is this same sensitivity not found in higher doses required for true anesthesia in contradiction to their previous work and that of others?

R2. We thank the reviewer for this question. From a PK point of view, older patients require less propofol during maintenance of anaesthesia to achieve the same PD effects of older patients. These relationships were described by Schnider et al, 1998-99. However, when performing an Entropy-guided anaesthesia, we note that there is the risk of an increasing in CeP targeting during surgery because of higher RE and SE values in the elderly caused by changes in the EEG (et al, 2020; PMID 32108685). In other words, Entropy (and, as demonstrated by other Authors, BIS (Ni et al, 2019; PMID: 31279479)) is not a reliable device to measure the different PD effects of propofol between old and young patients. In addition, the trial described by Ouattara et al., that we cited in the text, described the relationship between CeP and BIS, however they used another TCI model for propofol, the Marsh PK model, that does not account for age as a covariable. We used the Schnider TCI model instead, that also accounts for the age of the patient giving automatically the correct (for this model) dose of propofol to achieve a comparable CeP to the young and old, taking into account the different PK/PD of these populations. We stretched this concept in the manuscript.

Q3. Similarly, if older age is associated with increased sensitivity to anesthetics, why would the entropy values for wakefulness be higher in the elderly at comparable CeP? Doesn't this contradict the ROC data?

R3.Thanks for this comment! Please see our response to your previous question.

Q4 We have had significant experience with the BIS and entropy measures of consciousness and have always found a degree of EMG contamination, which the static entropy is "supposed" to filter out. Can the authors comment on this given that the patients were not paralyzed?

R4. Thank you for this comment. We completely agree with the reviewer that EMG provides a source for EEG signal contamination. Especially the frontal recording positions are prone to record EMG activity as well in awake and non-paralyzed patients. The frequency spectrum of the EEG and the EMG (under general anesthesia) overlap as for instance described by Kamata et al. (Kamata, Aho et al. 2011). Still, the major contamination occurs in the higher frequencies. Hence, as a rule of the thumb, a 30 Hz threshold to roughly separate signals dominated by EEG (<30 Hz) and EMG (>30 Hz) in non-paralyzed patients seems to work for anesthesia monitoring (Bonhomme and Hans 2007). The developers of the State (SE) and Response Entropy (RE) algorithm used this threshold to design two different indices. The SE processes information in the 0.8-32 Hz range and hence predominately the EEG, whereas the RE includes frequencies up to 47 Hz. This selection was made with a purpose, because RE is designed to quickly detect a potential arousal reaction and EMG activity increases during the arousal (Viertio-Oja, Maja et al. 2004). So in case of increasing EMG the SE and RE will start to drift apart because of an increased influence of high frequency oscillatory activity picked up by RE, but not SE. Another case, more important to our investigations is the age-related flattening of the power spectral density under general anesthesia (Kreuzer, Stern et al. 2020). This flattening also leads to an increased influence of higher frequencies that will also increase the difference between the RE and SE. So we are confident that the observed increase in the RE-SE with age under general anesthesia is not caused by increase EMG with age, but by the age-induced increased of higher EEG frequencies.

We stretched this concept in the manuscript.

Q5. How does this study differ from those mentioned in the following and why are the results so different:"CeP at loss of consciousness (LOC) and return of consciousness (ROC) during TIVA-TCI are lower in the elderly (age ≥65 years) than in youngers (age <65 years).[6,7] Further, it seems that lower CeP is necessary to maintain a similar anesthetic level in elderly using TIVA-TCI during cardiac surgery,[8] suggesting that older patients are more sensitive to Propofol administered for anesthesia maintenance."

R5. Thanks for this comment! Please see our response to your previous comment 2.

Federico: You may want to stretch this even more: While previous research clearly showed that a lower anesthetic dose is required in the elderly, the processed EEG indices like Entropy in our case are still higher. This clearly shows the potential risk of overdosing an old patient if you follow the index.

Q6. While formal PK/PD models for propofol and remifentanil have been established for quite some time, it is also well known that the variability in seeminly adequate effect site concentrations is quite high. Can the authors comment on this variability in their population?

R5. We thank the reviewer for this comment. We agree that the variability on adequate effect site concentration is quite high, however we tried to select a population as homogeneous as possible (women in menopause) and undergoing the same non-major surgery (breast surgery). However our results about the correlation between age and CeP at return of consciousness are in line with previous literature findings (Schnider et al, 1998-99). We comment about the variability of our population in the limitation section.

Q7. Can the authors comment more on their fentanyl doses as a confounding factor? Differential dosing here could grossly effect their results if not properly controlled. For instance, is the lack of difference in CeP and CeR between old and young patients during maintance of anesthesia possibly due to the younger patients receiving more fentanyl? They already state that such is the case in arousable patients: "Regarding the fentanyl dose, arousable patients received a significantly (p=0.027,AUC:0.26[0.19-0.47]) lower dosage of fentanyl..."

R7. Thank you for this comment. As presented in table1, the fentanyl doses as well as the intraoperative remifentanil doses did not differ significantly between the age groups. The AUC of 0.54 (Fentanyl) and 0.43 (CeR) did not indicate a relevant effect size as well. According to the traditional academic point system, AUC values between 0.4 and 0.6. 

We add this in the results section.

Q8. As I have no expertise in the statistics used, a more formal review of such by a statistician is warranted, especially in light of the significant scatter noted in the figures.

R8. Thanks for your honest response. We feel that our statistics used are appropriate to present our results. Reviewer 2 addressed some questions to the stats. Please see our response to his/her comments.

Q9. I am also confused by the conclusion that there is do difference in the CeR and CeP during maintenance, yet the entropy is higher in the elderly. This makes the older patients sound like they were merely not as deeply anesthetized compared to the younger patients at similar predicted Ce's and would actually suggest a decreased sensitivity to the anesthetic in the elderly!

R9. Thanks for this comment! Please see our response to your previous comment 2. The difference is due to the age-induced change in EEG features the EEG-based monitoring systems do not correct for.

Q10. On page 10, without referring to the figures, it is not clear which groups are being compared in the statements "When analyzing the pEEG indices, we found higher maximum (but not mean) RE (p<0.001;AUC:0.80[0.70-0.90]), SE (p=0.003; AUC:0.73[0.59-0.86]), RE–SE (p<0.001; 0.81[0.68-0.92])during maintenance (Figure 4D-F; Table 3). We also found higher maximum, but not mean, SPI values during maintenance (p=0.030; AUC:0.67[0.46-0.75]) (Figure S2, Table 3). There was a significant positive (p=0.006) correlation between maximum SPI and maximum RE–SE difference during surgery, i.e., max(RE-SE)=0.116*max(SPI)+1.969."

R10 We made changes as requested.

Q11. The authors' concerns regarding burst suppression are important ones. As such, they must comment on the presence or absence of burst suppression in all of the patients as this could give the very misleading result to which they refer, and will completely alter their interpretation of the EEG comparisons. I understand that they may not have the raw EEG data stored, but if most of the elderly are in fact more sensitive to anesthetics, thereby inducing burst suppression, this could be responsible for a titration to an SE and RE that are contaminated. If in burst suppression, the elderly might have needed far less anesthetic than given thereby altering their conclusions.

R11. Thanks for your thoughts on that and you are correct with your assumptions. This is indeed a complex issue and without having the raw EEG, we cannot give an ultimate answer. We are pretty certain that the 82-yr old, already mentioned had undetected BSupp because of the very high SE because this issue is known (Hart, Buchannan et al. 2009). Because BSupp detection focuses on the suppression phases (Särkelä, Mustola et al. 2002) and if contaminated they resemble an awake EEG and they will consequently cause high RE and SE values. Because the vast majority of our patients had indices in the index interval reflecting “adequate anesthesia”, i.e., between 40 and 60 (https://www.gehealthcare.co.uk/-/jssmedia/76841dd076a54dd5b1aa26e21c10e4cf.pdf?la=en-gb) . We are very confident that BSupp is not an issue in our data. 

R11. We stretched this question in the limit sections.

Q12 small corrections:

page 5: than in youngers, "youngers" is not a word.

page 5: helping do avoid should be helping to avoid

page 7: number of previous general anesthesia should read anesthetics

page 7: schooling’s years should read years of schooling

page 9: previous general anesthesia. should read anesthetics

R12. We made changes as requested.

Reviewer #2:

 Some aspects of the statistical analyses had me a bit confused. I have made some comments below in an attempt to highlight some of the issues. Generally speaking, the analyses are pretty straightforward.

Q1. The abbreviation AUC is not defined anywhere in the manuscript. If this happens to be area under the ROC curve, this is not mentioned in the methods, I don't understand the usage in this manuscript, and I recommend cutting it unless you can provide justification. It also makes the text of the results section too number-heavy and I would consider removing from the text regardless.

R1. Thank you for this comment. We made changes as requested.

Q2. The biggest problem I had with the analyses was the switching between assumptions for the outcome variables. Median and IQR summary statistics and Wilcoxon are used when the normality assumption is suspect. Linear regression and Pearson correlation coefficients are used when a normality assumption is permissible to make. For instance, in section 3.2, in the first paragraph, CeP and CeR are all reported as median IQR (with p-values from Mann-Whitney, if I understood correctly). This switches in the second paragraph to assuming normality with for the linear model, and then back to a nonparametric test for testing young v. old. Neither are necessarily wrong, but the consistency has me baffled. Maybe more explanation for these choices in the methods section would be helpful.

R2. Thank you for this comment. We highly appreciate your thoughts on this. We tried to keep the stats as simple as possible. In order to perform the group comparison, we decided to run the MWU tests because testing for normality and heteroscedasticity is not necessary. And testing for normality is always tricky itself. We agree that the t-stat for the linear model comes from a parametric test that considers the standard error. This is part of MATLB fitlm algorithm. We are aware that the results from the parametric linear model testing and the nonparametric group testing are mostly redundant and we can understand your confusion about it. Hence, we decided (as you also suggest in comment #3) to only show the slopes, intercepts and R2 for our linear models and leave the inference stats to the grouped comparisons that we used the MWU-test for.

Q3. I also found it very confusing that both a linear model and a Pearson correlation were used. There are differences between the two (mainly that both variables in correlation are considered to be sampled with error, whereas only the outcome (Y) variable is in a linear model), but there is nothing in the methods to support the use of both. Personally, I would report the slopes of the linear regression models. Intercepts are sometimes not reported so I will defer to you as to whether that is needed.

R3. Thanks for this comment! Please see our response to your previous comment.

Q4. Did you consider fitting multivariable regression models to account for any potential confounders?

R4. Thanks for this comment. We did not in the first version, because our sample size is rather small to fit a multivariate regression model. One of our main conclusions is that SE and RE seem to increase with age during adequate levels of anesthesia. Hence, we performed a small multivariate regression analysis of the type:

SE ~ 1 + age + CePmean + CeRmean

RE ~ 1 + age + CePmean + CeRmean

RE-SE ~ 1 + age + CePmean + CeRmean

As you can see in the results below, age remains the driving factor.

SE ~ 1 + age + CePmean + CeRmean 

 Estimate SE tStat pValue 

 ________ _______ ______ ________ 

 (Intercept) 12.268 9.3506 1.312 0.19486

 age 0.34617 0.11042 3.1349 0.002737

 CePmean 2.8489 2.0308 1.4028 0.1662

 CeRmean 2.964 2.6745 1.1082 0.2725

RE ~ 1 + age + CePmean + CeRmean 

 Estimate SE tStat pValue 

 ________ _______ _______ _________ 

 (Intercept) 9.4297 10.21 0.92361 0.35965

 age 0.40835 0.12057 3.3869 0.0012989

 CePmean 3.3563 2.2174 1.5136 0.13575

 CeRmean 3.8975 2.9202 1.3347 0.18739

RE-SE ~ 1 + age + CePmean + CeRmean 

 Estimate SE tStat pValue 

 ________ _______ _______ ________ 

 (Intercept) -2.8386 1.8325 -1.5491 0.127

 age 0.062187 0.02164 2.8736 0.005724

 CePmean 0.50744 0.39799 1.275 0.20757

 CeRmean 0.93354 0.52414 1.7811 0.080319

Q5. In the figures, please ensure that the y-axis range for the scatter plot matches the y-axis range for the boxplots.

R5. We made changes as requested. 

Q6. Also, I suggest expanding the y-axis to zero for all plots, e.g, Figure 2 A.

R6. We made changes as requested. 

Reviewer #3: 

I am quite impressed with the methodological approach that you took, and the care that was displayed in its execution.

Using exacting methods you quantified an important domain of clinical practice and have brought illumination to a foggy domain that was sorely in need of scientific foundation.

That one of the investigators was involved in hands-on data acquisition was a bit troubling and I believe you need to demonstrate absence of potential bias in their data recording.

The statistical analysis was appropriate in all regards.

The data capture excellent and sample size more than ample.

My overall recommendation is to accept this fine work of science that has virtually immediate clinical relevance and application.

Thank you for your comments!

6. PLOS authors have the option to publish the peer review history of their article (what does this mean?). If published, this will include your full peer review and any attached files.

Do you want your identity to be public for this peer review? For information about this choice, including consent withdrawal, please see our Privacy Policy.

Reviewer #1: No

Reviewer #2: No

Reviewer #3: Yes: CJ Biddle

References

Bonhomme, V. and P. Hans (2007). "Muscle relaxation and depth of anaesthesia: where is the missing link?" British Journal of Anaesthesia 99(4): 456-460.

Hart, S. M., C. R. Buchannan and J. W. Sleigh (2009). "A failure of M-Entropy to correctly detect burst suppression leading to sevoflurane overdosage." Anaesth Intensive Care 37(6): 1002-1004.

Kamata, K., A. Aho, S. Hagihira, A. Yli-Hankala and V. Jäntti (2011). "Frequency band of EMG in anaesthesia monitoring." British journal of anaesthesia 107(5): 822-823.

Kreuzer, M., M. A. Stern, D. Hight, S. Berger, G. Schneider, J. W. Sleigh and P. S. García (2020). "Spectral and Entropic Features Are Altered by Age in the Electroencephalogram in Patients under Sevoflurane Anesthesia." Anesthesiology 132(5): 1003-1016.

Särkelä, M., S. Mustola, T. Seppänen, M. Koskinen, P. Lepola, K. Suominen, T. Juvonen, H. Tolvanen-Laakso and V. Jäntti (2002). "Automatic Analysis and Monitoring of Burst Suppression in Anesthesia." Journal of Clinical Monitoring and Computing 17(2): 125.

Viertio-Oja, H., V. Maja, M. Sarkela, P. Talja, N. Tenkanen, H. Tolvanen-Laakso, M. Paloheimo, A. Vakkuri, A. Yli-Hankala and P. Merilainen (2004). "Description of the Entropy algorithm as applied in the Datex-Ohmeda S/5 Entropy Module." Acta Anaesthesiol Scand 48(2): 154-161.

---

## [Decision Letter · Decision Letter 1]

4 Dec 2020

Age influences on Propofol estimated brain concentration and Entropy during maintenance and at return of consciousness during Total Intravenous Anesthesia With Target-Controlled Infusion in Unparalyzed Patients: an Observational Prospective Trial

PONE-D-20-16382R1

Dear Dr. Linassi,

We’re pleased to inform you that your manuscript has been judged scientifically suitable for publication and will be formally accepted for publication once it meets all outstanding technical requirements.

Kind regards,

JianJun Yang, M.D., Ph.D.

Academic Editor

PLOS ONE

Additional Editor Comments (optional):

Reviewers' comments:

Reviewer's Responses to Questions

**Comments to the Author**

1. If the authors have adequately addressed your comments raised in a previous round of review and you feel that this manuscript is now acceptable for publication, you may indicate that here to bypass the “Comments to the Author” section, enter your conflict of interest statement in the “Confidential to Editor” section, and submit your "Accept" recommendation.

Reviewer #1: All comments have been addressed

Reviewer #2: All comments have been addressed

2. Is the manuscript technically sound, and do the data support the conclusions?

Reviewer #1: Yes

Reviewer #2: (No Response)

3. Has the statistical analysis been performed appropriately and rigorously? 

Reviewer #1: N/A

Reviewer #2: (No Response)

4. Have the authors made all data underlying the findings in their manuscript fully available?

Reviewer #1: Yes

Reviewer #2: (No Response)

5. Is the manuscript presented in an intelligible fashion and written in standard English?

Reviewer #1: Yes

Reviewer #2: (No Response)

6. Review Comments to the Author

Reviewer #1: (No Response)

Reviewer #2: (No Response)

7. PLOS authors have the option to publish the peer review history of their article (what does this mean?). If published, this will include your full peer review and any attached files.

Reviewer #1: No

Reviewer #2: No

---

## [Editor Report · Acceptance letter]

9 Dec 2020

PONE-D-20-16382R1 

Age influences on propofol estimated brain concentration and entropy during maintenance and at return of consciousness during total intravenous anesthesia with target-controlled infusion in unparalyzed patients:an observational prospective trial 

Dear Dr. Linassi:

I'm pleased to inform you that your manuscript has been deemed suitable for publication in PLOS ONE. Congratulations! Your manuscript is now with our production department. 

Kind regards, 

on behalf of

Dr. JianJun Yang 

Academic Editor

PLOS ONE